# Tensile Properties of Curaua–Aramid Hybrid Laminated Composites for Ballistic Helmet

**DOI:** 10.3390/polym14132588

**Published:** 2022-06-26

**Authors:** Natalin Michele Meliande, Pedro Henrique Poubel Mendonça da Silveira, Sergio Neves Monteiro, Lucio Fabio Cassiano Nascimento

**Affiliations:** Department of Materials Science, Military Institute of Engineering-IME, Praça General Tibúrcio, 80, Urca, Rio de Janeiro 22290-270, Brazil; nmeliande@gmail.com (N.M.M.); snevesmonteiro@gmail.com (S.N.M.); lucio@ime.eb.br (L.F.C.N.)

**Keywords:** natural fiber, aramid, curaua, hybrid composite, ballistic helmet, tensile test, ballistic armor

## Abstract

A typical ballistic protection helmet for ground military troops has an inside laminate polymer composite reinforced with 19 layers of the aramid, which are neither recyclable or biodegradable and are relatively expensive. The hybridization of synthetic aramid with a natural lignocellulosic fiber (NLF) can provide a lower cost and desirable sustainability to the helmet. In the present work, the curaua fiber, one of the strongest NLFs, is, for the first time, considered in non-woven mat layers to partially replace the aramid woven fabric layers. To investigate the possible advantage of this replacement, the tensile and impact properties of aramid/curaua hybrid laminated composites intended for ballistic helmets, in which up to four layers of curaua were substituted for the aramid, were evaluated. Tensile strength, toughness, and elastic modulus decreased with the replacement of the aramid while the deformation of rupture was improved for the replacement of nine aramid layers by two layers of curaua. Preliminary impact tests corroborate the decreasing tendency found in the tensile properties with the replacement of the aramid by curaua. Novel proposed Reduction Maps showed that, except for the replacement of four aramid layers by one layer of curaua, the decrease percentage of any tensile property value was lower than the corresponding volume percentage of replaced aramid, which revealed advantageous hybridization for the replacement of nine or more aramid layers.

## 1. Introduction

As a general class of materials, composites have been the most successful, since the past century, for their unsurpassed advantages of relative lightness, improved mechanical properties, durability, and fatigue resistance, among others related to the engineered combination of distinct materials with stronger interfacial bonding.

In particular, fiber-reinforced polymer composites are currently indispensable and extensively applied in food packing, sports goods, automobile parts, and ballistic armors for personal protection. The performance of these composites might be enhanced by hybridization of the reinforcement with different fibers, which could be either synthetic or natural. A typical case, to be explored in the present work, is the use of hybrid synthetic/natural fibers composites in ballistic helmets.

A US-developed helmet for head protection within the American armor system for ground troops (PASGT) has, in the past 30 years, been widely adopted [1,2], including by the Brazilian army [3]. The PASGT is manufactured by stacking 19 layers of a 76 vol% aramid fabric reinforced polymer composite with a thickness varying from 8 to 10 mm [2,3], as illustrated in Figure 1. Although they display superior mechanical properties, as any synthetic fiber, the aramid, and neither recyclable nor biodegradable material, constitute a storage problem due to their mandatory disposal after 5 years of use [3]. In addition, aramid fabrication is associated with CO_2_ emissions and high cost [4,5]. Hybridization with synthetic fibers, such as carbon and glass, might meliorate the mechanical properties [6,7,8] and slow synergetic effects [6], mainly for structural constructions. However, hybrid synthetic-only fibers are not a solution for the aforementioned problems.

In principle, hybridization of the aramid with natural lignocellulosic fibers (NLF) would be a promising option for application in helmets. NLFs are obtained from plants which are renewable and abundant resources, associated with low energy production requirements. As NLFs are biodegradable and include CO_2_ emission-free processing, they are considered environmentally friendly. Many of these fibers are native or grown in developing regions and represent an important income source for local populations [9]. Moreover, compared to synthetic fibers, NLFs also pose less health risk during processing, and they are sustainable [10,11,12,13,14,15].

Among the NLFs, the curaua fiber stands out. The curaua plant (*Ananas erectifolius*) is a native species from the Brazilian Amazon region from which fibers with superior mechanical properties, known for a long time by local indigenous people, are extracted from leaves. They use curaua fibers to produce ropes, hammocks, fishing lines, and tools that require high tensile strength and deformation capacity. As shown in Table 1, the curaua fibers display low density and high tensile strength, compared with synthetic fibers, resulting in an exceptional specific strength [16,17,18], as presented in Table 1.

On the other hand, NLFs such as curaua are considered challenging materials, since they are not free from defects and are microstructurally heterogeneous. These two typical NLF characteristics are the main cause for the NLFs’ huge property variation compared to synthetic fibers, which are uniformly fabricated and free from defects [19]. Moreover, the mechanical properties of NLFs from the same species can vary considerably depending on the age of the plant and the crop location, as well as climatic and soil conditions [19,20,21,22].

In addition to these characteristic shortcomings, the NLF–matrix adhesion tends to be weak, impairing the composite’s overall mechanical performance. It happens because of the NLFs’ hydrophilicity. The water on the fiber surface impairs its functional interaction with a hydrophobic polymer, and prevents the fiber wetting by the matrix [19]. In this context, the hybridization of NLFs with synthetic fibers in polymer matrix composites can provide a balance between low production cost and environmental problems, as well as satisfactory mechanical properties [23]. This is because in hybrid composites, the disadvantages of one type of fiber could be compensated for the advantages of the other. As a result, more sustainable materials with a reasonable production cost and acceptable mechanical properties can be obtained. Thus, many research works are investigating the possibility of using NLFs and synthetic fiber hybrid composites for technical applications [23,24,25,26,27,28,29,30,31,32,33,34,35,36,37,38,39,40,41,42,43,44,45,46,47,48,49,50].

Despite the large amount of research on plain NLF-reinforced composites for ballistic application [51,52,53,54,55,56,57,58,59,60,61,62,63,64,65,66,67,68,69,70,71,72], the most promising use of these fibers in ballistic composites is together with synthetic fibers [26,30,34,35,41,42]. For structural and liability applications, such as in ballistic protection, composite materials must exhibit durability and reliability in performance. Furthermore, plain NLF-reinforced composites have a lower impact resistance than traditional ballistic composites. The combination of NLF with synthetic fibers could overcome these typical shortcomings. At the same time, the natural fiber substitution for synthetic ones could reduce the environmental problems from the production of traditional ballistic fibers and the disposal of the composites, not to mention the significantly lower production cost [52].

As the laminated composite ballistic performance is attributed to its deformation capacity in the transverse direction, and its interlaminar shear strength, the greater this deformation capacity and the level of delamination, the greater the capacity to absorb the impact energy. Based on these premises, it is important to know the material’s mechanical properties, especially its tensile strength, deformation at the rupture, and toughness, in order to qualitatively estimate the material’s ballistic potential.

With the intention of future helmet development, the aim of the present work is to determine the tensile properties of curaua–aramid hybrid laminated composites in order to evaluate the hybridization effect for possible ballistic application.

## 2. Materials and Methods

### 2.1. Materials

For the polymeric matrix of the hybrid composites proposed in this work, commercial epoxy resin of the diglycidyl ether, of the bisphenol A (DGEBA), hardened with triethylenetetramine (TETA) in a ratio of 13 per 100, was used. In addition to lower cost, the epoxy resin was chosen due to its ease of acquisition, storage, and room temperature processing, in contrast to PVB-phenolic film, commonly used in ballistic helmets.

Aramid woven fabric T750, produced by Teijin Aramid (Arnhem, The Netherlands), was used in the laminated composites. With 3360Dtex-linear density Twaron^®^1000 yarns and a 460 g/m^2^ areal density, this synthetic woven fabric is commonly used in ballistic helmets. Curaua non-woven mat fibers with about 26 g/m^2^ of areal density were the NLF hybrid counterpart precursor. It was produced by a Brazilian company, Pematec Triangel do Brasil (Santarém, Brazil). These precursor materials are illustrated in Figure 2.

### 2.2. Composites Production

An important point, directly related to hybrid composite configuration that is proper of the present work, is the number of layers corresponding to the distinct synthetic and natural fiber reinforcements. As illustrated in Figure 1, the traditional PASGT ballistic helmet is fabricated by stacking 19 layers of aramid fabric composite. As such, the number of composite layers associated with a final thickness stack of 8–10 mm [2,3] is the main parameter for a density between 10 to 12 kg/m^2^ corresponding to the stack weight of approximately 200 g, which is important for the combatant wearer’s comfort. The substitution of curaua mat for aramid fabric layers must keep these same thickness and density parameters.

Therefore, the first step was to find the number of curaua mat layers that could completely replace the 19 layers of aramid fabric, still preserving the 8–100 mm thickness and 10–12 kg/m^2^ areal density for the stack. Based on preliminary experiments, this number for a curaua mat was only found as four layers, which gives 9.98 mm and 10.69 kg/m^2^ for stack thickness and density, respectively. Then, different hybrid stacks were fabricated comprising whole numbers: 1, 2, and 3 of the curaua mat layers. Proportionally these hybrid stacks would have 15, 10, and 5 layers of aramid fabric, as presented in Table 2 with the corresponding denoted configuration.

Table 3 shows the composite characteristics, including those of actual 19 aramid layers of a PASGT-based composite. As can be seen for all configurations, the characteristic variability was very small. Although resin quantities were calculated considering around a 60 vol% composite reinforcement, the produced composites do not present the same values. It can be seen in Table 3 that the greater the aramid layers number, the greater the composite reinforcement volume fraction. This can be explained by the perfectly smooth aramid fiber surface, which hinders its wettability by the resin, as opposed to the imperfect curaua fiber surface which facilitates its wettability. In addition, the aramid woven fabric weft is extremely closed, which also hinders the wettability by the resin, unlike the curaua non-woven material, in which there are voids and discontinuities through which the resin can penetrate [56,60].

For the composites production, a steel mold with dimensions of 150 × 120 × 11.9 mm was used. Before production by hand lay-up—Figure 3—the curaua non-woven mat was dried at 70 °C for at least 24 h, in order to reduce its moisture and, therefore, enhance the epoxy–curaua interfacial adhesion [17,60]. The press, shown in Figure 3c, applied a pressure of 5–7 t, about 4 MPa, during cold compression molding, for at least 8 h, as specified by the epoxy manufacture for resin curing time. In spite of applied pressure, the relatively high viscosity of the epoxy is associated with a fluidity, which causes difficulties for the resin to infiltrate both the aramid fabric and the curaua mat well. Ongoing works are investigating the method of vacuum infusion to ensure the effective removal of bubbles from the epoxy matrix and its fibers interface. All investigated composites were produced in a non-alternating configuration since alternating a synthetic and natural fiber layers configuration might cause inferior mechanical and ballistic properties [35]. The E-10A/2C composite production is exemplified in Figure 3.

### 2.3. Uniaxial Tensile Test

Quasi-static uniaxial tensile tests were performed in accordance with ASTM D3039/D3039M-17 [73]. In order to obtain the tensile stress-strain curves and to determine the relevant mechanical properties, 12 specimens of each type of composite were tested. The specimens had a constant rectangular cross-section, with a 150 mm length, 19 mm width, and thickness, according to Table 3, and a 50 mm gage length. Tensile tests were performed in an Instron machine with a 2 mm/min crosshead speed, shown in Figure 4. The specimens were held at both ends and covered with non-slip abrasive paper by press-controlled grips, as seen in Figure 4c. Applied force was registered by the machine load cell, and the elongation measured by the extensometer is also shown in Figure 4c.

### 2.4. Impact Test

Preliminary Charpy impact tests were carried out in single notched samples for the five composites configurations investigated—Table 2—in order to provide a basis for the ongoing works on ballistic resistance of helmets. These Charpy tests followed the ASTM D 6110-18 standard and were conducted in a Tinus Olsen Impact 104 electronic instrumented hammer machine.

## 3. Results and Discussion

### 3.1. Tensile Properties

Figure 5 shows the composites’ stress-strain curves obtained in the tensile tests, illustrated in Figure 4c. It can be observed that, for all investigated composites, the curve has an initial linear region as shown in the graph’s magnified insets. This region corresponds to the material linear elastic regime and is followed by a non-linear region up to the fracture, corresponding to the non-linear elastic regime. Similar curves were obtained by Salman et al. [36], from the tensile test of a Kevlar^®^/kenaf woven fabric hybrid laminated composite with a 35 wt.% epoxy.

Composite ultimate tensile strength (UTS), deformation at the rupture (ε_rup_), elastic modulus, and tensile toughness are shown in Table 4. For each composite, the value of the elastic modulus was obtained from the linear fit slope of the initial curve region. The value of toughness was inferred from the area under the stress-deformation curve up to the fracture. Initially, analyzing the results presented in Table 4, less dispersion tendency can be noticed as the number of aramid layers decreases while that of curaua increases. This is corroborated by the decreasing values of standard deviations from E-19A/0C to E-0A/4C, from E-5A/3C to E-0A/4C. However, when analyzing the properties among E-0A/4C specimens extracted from the same composite plate, i.e., from CP_01 to CP_06 (E-0A/4C*) and from CP_07 to CP_12 (E-0A/4C**), it is clear in Table 4 that standard deviations consistently decreased from E-5A/3C to E-0A/4C. For the E-0A/4C composite, the disparity between mechanical properties from one plate to another can be justified by the NLFs’ inherent property variability, as widely discussed in the literature [16,20,21,22].

The homogeneity in dispersion associated with aramid layer reduction and curaua layers increases, and can be explained by these materials’ structure types, i.e., woven fabric and non-woven mat, respectively. This is because, in the aramid woven fabric case, the yarns are orthogonally oriented in weft. Thus, depending on tissue cut and its positioning in the laminate, the preferred orientations may undergo small changes, which might affect the tensile strength. On the other hand, in the curaua non-woven mat, fibers are randomly oriented, which gives an isotropic character to the composite.

Based on Table 4, Figure 6 illustrates that UTS decreases when replacing aramid layers with curaua layers. This was already expected due to the aramid’s superior mechanical properties in relation to NLFs [10,16]. However, it should be noted that there is no reduction in UTS from composites E-15A/1C to E-10A/2C. In order to corroborate this observation and verify whether there are significant differences between the composites of UTS, an analysis of variance (ANOVA) was carried out with a 5% significance level.

From ANOVA results of F >> F_critic_, shown in Table 5, it can be stated with 95% confidence that composites have different UTS. In order to compare the composites’ UTS to each other, a Tukey’s test was applied based on the minimum significant difference (MSD) that must exist between two UTSs, so that they are significantly different with a 5% significance level. All the composite UTS differences were greater than MSD, except for E-15A/1C and E-10A/2C composites. This confirms the preliminary observation from Figure 6.

Based on Table 4, it is shown that there are no significant variations in ε_r_ between E-19A/0C, E-10A/2C, and E-5A/3C composites. However, for both E-15A/1C and E-0A/4C, there seems to be a significant reduction on ε_r_ compared to the others. Thus, in order to corroborate these observations and verify whether there are significant differences between this mechanical property of the composites, ANOVA was carried out with a 5% significance level.

From ANOVA results, shown in Table 6 of F >> F_critic_, it can be stated with 95% confidence that the ε_r_ are different. In order to compare these composites performed between each other, Tukey’s test was applied based on the MSD that must exist between two ε_r_, so that they are significantly different with a 5% significance level. The differences between composites’ E-19A/0C, E-10A/2C and E-5A/3C ε_r_ are smaller than the MSD. This confirms the preliminary observation that these composites’ ε_r_ are similar, despite the lower relative aramid content in E-10A/2C and E-5A/3C composites. In addition, it can also be observed that the differences between these composites’ ε_r_ and those of E-15A/1C and E-0A/4C composites are greater than MSD. This corroborates the initial observation that there was a significant reduction in ε_r_ for E-15A/1C and E-0A/4C composites.

Based on Table 4, it is shown that the tensile toughness tends to decrease with aramid layer reduction. Despite this, it should be noted that there was no significant variation in this property from E-15A/1C to E-10A/2C composites. In order to corroborate these observations and verify if there are significant differences between the composites toughness’, an ANOVA was carried out with 5% significance level.

From the ANOVA results of F >> F_critic_, shown in Table 7, it can be stated with 95% confidence that there are different toughness levels. In order to compare the composite’s properties to each other, a Tukey’s-test-based MSD was applied that must exist between two toughness levels so that they are significantly different with a 5% significance level. The composites’ toughness differences were greater than MSD, except for those of E-15A/1C and E-10A/2C composites. This confirms the preliminary observation that their toughness levels are similar, despite the lower relative aramid amount in the E-10A/2C composite.

Based on Table 4, it is shown that the elastic modulus decreased with aramid layer reduction. Despite this, it should be noted there was no significant variation from E-19A/0C to E-15A/1C composites. In order to corroborate these observations and verify if there are significant differences between the composites’ elastics modulus, an ANOVA was carried out with a 5% significance level.

From the ANOVA results, shown in Table 8, of F >> F_critic_, it can be stated with 95% confidence that there are different elastic moduli. In order to compare these composites’ properties to each other, a Tukey’s test based on the MSD was applied, which must exist between two elastic moduli, so that they are significantly different with a 5% significance level. The composite differences were greater than MSD, except for those of E-19A/0C and E-15A/1C composites, as well as E-5A/3C and E-0A/4C composites. This confirms the preliminaries observations that the value of the elastic modulus decreased with aramid layer reduction, and that the E-19A/0C and E-15A/1C elastic modulus values are similar. Furthermore, this test indicated that there is no significant difference between E-5A/3C and E-0A/4C elastic moduli.

### 3.2. Reduction Maps

A preliminary quantitative analysis on the effect of reducing the aramid layers will provide support to discuss the tensile properties’ gain or loss caused by incorporation of curaua layers. In order to perform this analysis, a graphical representation by means of “Reduction Maps” was proposed, which provides a view of the comparative decrease in each property value against the reduction in the volume of the aramid. These Reduction Maps are constructed from volumes listed in Table 3 and values of tensile properties in Table 4.

Based on data in Table 4, graphs of the percentage reduction in the property value for a given composite are plotted versus the percentage reduction in the volume of the aramid being replaced by curaua in the composite. Figure 7 shows the Reduction Maps for the distinct tensile properties investigated. These maps plot the property reduction value for each composite, both for the main value (red solid square) and lower limit considering the standard derivation (red open circle). Maps in Figure 7c,d also show green points below the 0% horizontal line, which correspond to an increase, rather than reduction, in property caused by the aramid replacement.

All graphs in Figure 7 display a dashed straight line from 0% to 100%, dividing an upper zone, called “loss”, associated with reduction values of the property greater than the corresponding volume reduction of the aramid. In contrast, the lower “gain” zone corresponds to proportionally advantageous reduced values, smaller than the volume reduction of the aramid. It is noteworthy that only the UTS and toughness mean values of composite E-15A/1C (replacing four layers of the aramid by one layer of curaua), in Figure 7a,b, respectively, disclose proportionally disadvantageous substitutions. In other words, the composite loses almost 40% of its ultimate strength and 47% of tensile toughness by replacing, volume-wise, 29% of the aramid for curaua. Salman et al. [36] carried out uniaxial tensile tests with a Kevlar^®^/kenaf woven fabric hybrid laminated composite with an epoxy matrix, which can be considered equivalent to the E-15A/1C composite. The authors obtained for their composite about 11 GPa of the elastic modulus, similar to E-15A/1C. Furthermore, these authors obtained about 200 MPa for UTS and 3.40% for ε_r_, also similar to the E-15A/1C composite. This confirms the validity of the present results.

On the contrary, all other replacements of the aramid by curaua are advantageous since the mean and lower limit points fall in the gain zone. Moreover, the replacement of 10 aramid layers by 2 curaua layers, for composite E-10A/2C, might be improved by almost 20% of the deformation to the rupture, as shown in Figure 7c.

The results in Table 4 and Figure 7 revealed that, except for composite E-15A/1C, the other replacements of the aramid by curaua, up to 100%, present advantages in terms of relative gain in mechanical properties. This serves as an incentive to investigate the ballistic performance of helmets made with these hybrid aramid/curaua composites. In particular, the composite E-10A/2C appears to be the optimal hybrid configuration supported by the results in Table 4 and Figure 7.

### 3.3. Impact Test Results

Table 9 presents the Charpy impact test preliminary single sample results for the investigated composites. In this table, one should notice the decrease in absorbed impact energy with the amount of curaua mat layers. This behavior follows the same trend presented in Table 4 and Figure 6 for the UTS from tensile tests, and corroborates the role-played by the hybridization of epoxy composites with aramid and curaua fibers.

### 3.4. Tensile Rupture Mechanisms

Macroscopic analysis of the specimen failure modes after the tensile test justifies the trend observed in Figure 6 and Table 4. As seen in Figure 8a, the E-19A/0C specimen without curaua suffered delamination from the outermost to the innermost aramid layers. Moreover, aramid yarn rupture occurred in the outermost layers; see Figure 8a. By introducing one curaua non-woven mat layer (E-15A/1C composite), the specimen suffered curaua fiber pullout and rupture; see Figure 8d. It is important to mention that the non-woven mat consists of short fibers randomly distributed in layers which are mechanically stacked and compacted. In such a case, fiber pullout possibly occurs more frequently than fiber rupture.

This, however, happens long before the more intense aramid layer delamination and the aramid yarn rupture, observed in the E-19A/0C specimen. This can be explained in terms of the lower curaua/epoxy layer resistance. First, this is because the non-woven mat does not have the woven fabric structure; second, because fiber pullout is usually easier than fiber rupture; and third, because NLF resistance is normally inferior [16].

The introduction of one more curaua layer reduces the aramid amount and, therefore, creates a barrier to crack propagation, due to the layers’ interface causing delamination. The net result is a marginal composite UTS reduction (Table 4) associated with aramid layer delamination and aramid yarn rupture intensification, as shown in Figure 8c. It is important to mention that there is practically no delamination between curaua layers. This corroborates the hypothesis that, owing to the non-woven mat structure, the curaua/epoxy layer behaves as a more homogeneous and isotropic material, and its fracture seems to be predominantly ductile. Indeed, Gupta and Srivastava [23] showed that plastic deformation on natural fiber-reinforced polymer composites increases due to the incorporation of comparably high elongation fibers.

The net effect of introducing one more curaua layer (E-5A/3C composite) to replace another five aramid layers causes a significant UTS reduction. This is because all the aramid in the 10T2C composite was already “working” in its entirety, and the replacement of five aramid layers by one curaua one did not pay off. Thus, there was a marked intensification of the aramid failure mechanisms, such as aramid layer delamination and aramid yarn rupture. In fact, as indicated by Rahman et al. [24], the different enhancement effects in the aramid-reinforced composites might be attributed to interfacial adhesion with the polymer matrix. On the other hand, the curaua failure occurs basically from fiber pullout and rupture, as shown in Figure 8d. Finally, the composite E-0A/4C UTS decreased considerably with the total aramid replacement, confirming the mechanical properties’ superiority of the aramid woven fabric in relation to the curaua non-woven mat. In this case, the fracture occurred entirely due to curaua fiber pullout and rupture, as shown in Figure 8e.

## 4. Summary and Conclusions

Tensile properties and preliminary impact results of laminated hybrid composites, intended for ballistic helmets, in which traditional aramid fabric layers were combined with a curaua non-woven fiber mat, allowed conclusion of the following.

Except for the tensile deformation at the rupture, the ultimate stress (UTS), elastic modulus, toughness, and absorbed impact energy decreased with an increasing amount of curaua substitution for aramid layers.Proposed Reduction Maps revealed a percentage loss in UTS (~40%) and toughness in volume percentage (29%) of the aramid, which corresponds to a disadvantageous substitution of four layers of the aramid by one layer of curaua.On the contrary, the percentage of reduction in all tensile properties for the substitution of 9 and 14 layers of the aramid by, respectively, 2 and 3 layers of curaua was found advantageous. In particular, the composite with a ~33 vol.% of the aramid (10 layers) and ~32 vol.% of curaua (2 layers) in a ~35 vol.% epoxy matrix were the most promising for ballistic helmet.The absence of a delamination mechanism in the rupture of curaua layers contributes to an increase in deformation to the rupture, associated with a more plastic hybrid composite and possibly an improved ballistic performance

## Figures and Tables

**Figure 1 polymers-14-02588-f001:**
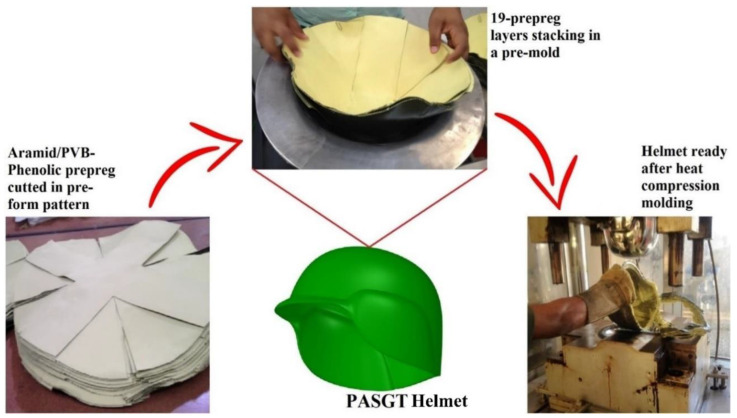
Thermoset ballistic helmet production process.

**Figure 2 polymers-14-02588-f002:**
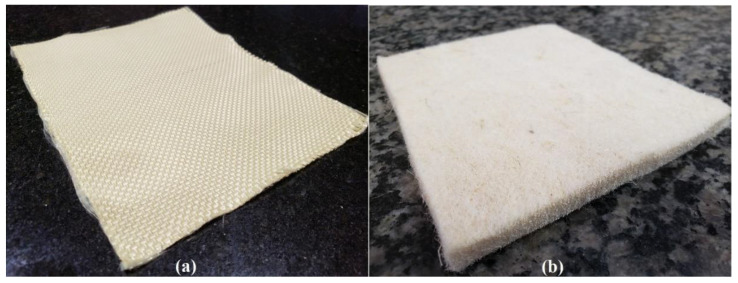
(**a**) Aramid woven fabric and (**b**) curaua non-woven mat.

**Figure 3 polymers-14-02588-f003:**
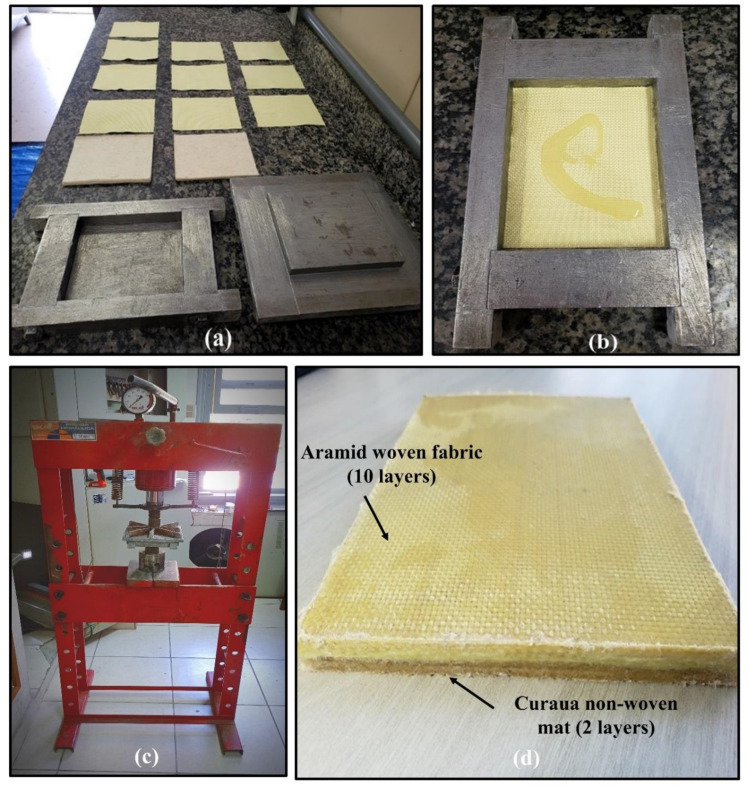
E-10A/2C composite production process: (**a**) aramid woven fabric, curaua non-woven material, and metallic mold besmeared with a release agent ready for cold molding; (**b**) epoxy over aramid woven fabric during processing; (**c**) cold compression molding, and (**d**) fabricated E-10A/2C composite plate.

**Figure 4 polymers-14-02588-f004:**
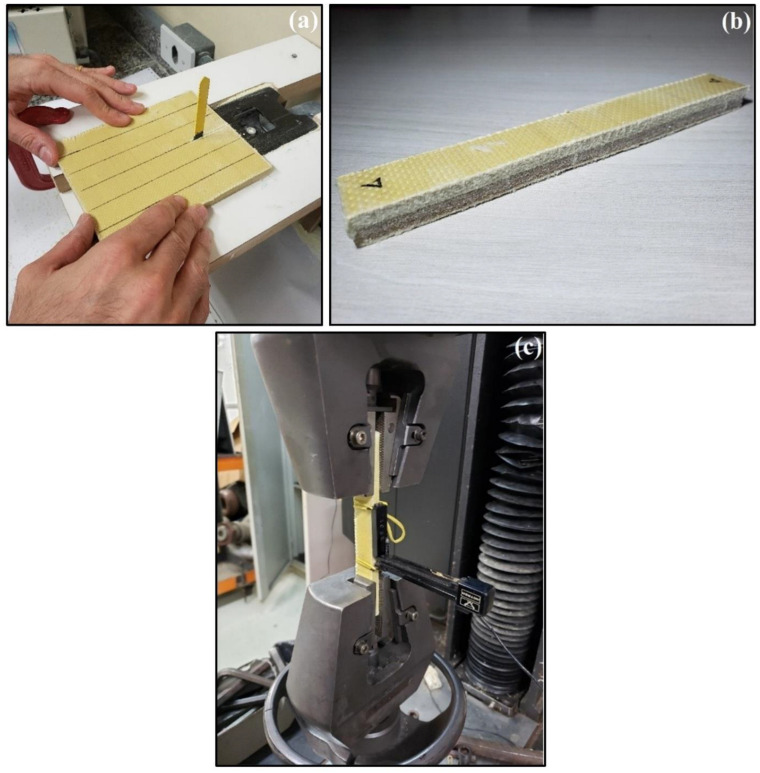
(**a**) Tensile test specimens cutting; (**b**) tensile test specimens ready; and (**c**) composite specimen tensile being tested.

**Figure 5 polymers-14-02588-f005:**
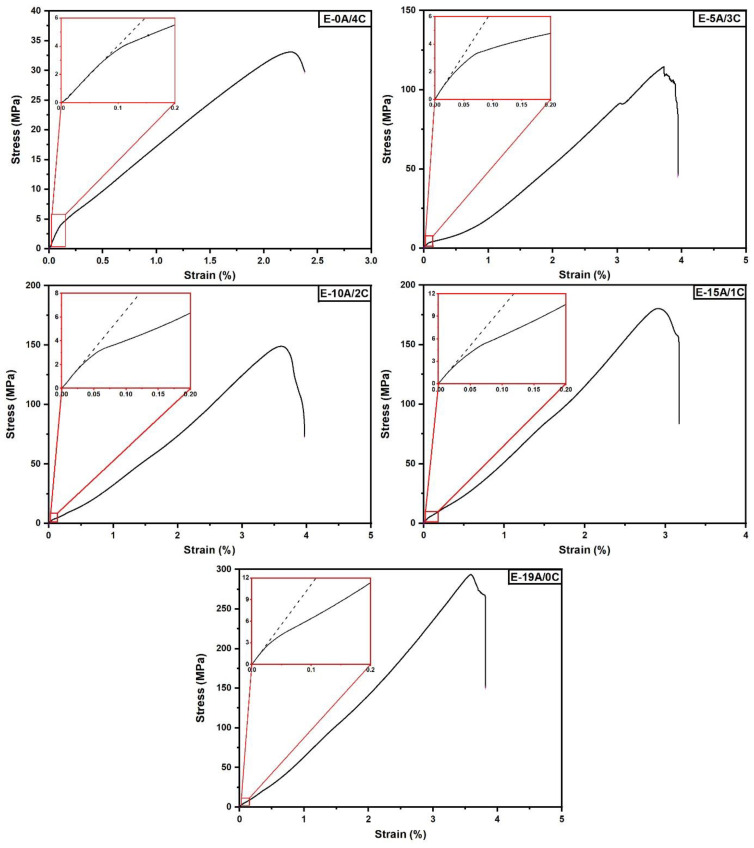
Typical stress-strain (σ-ε) curves obtained from the composites’ tensile tests.

**Figure 6 polymers-14-02588-f006:**
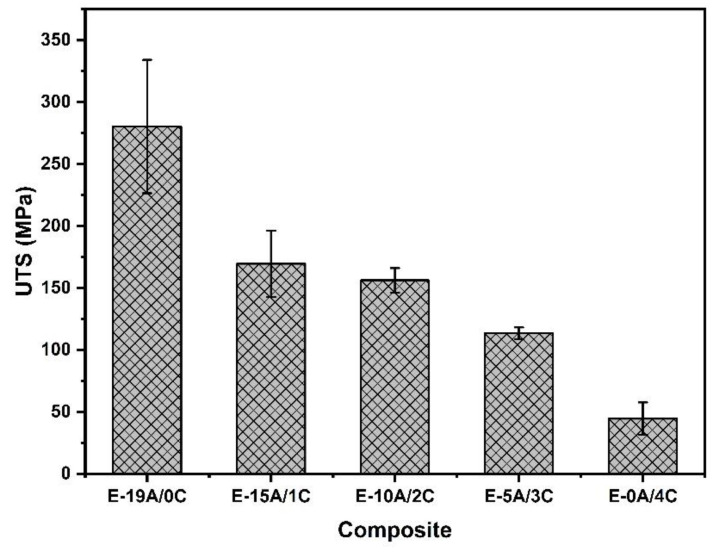
Comparison between composites UTS obtained from tensile tests.

**Figure 7 polymers-14-02588-f007:**
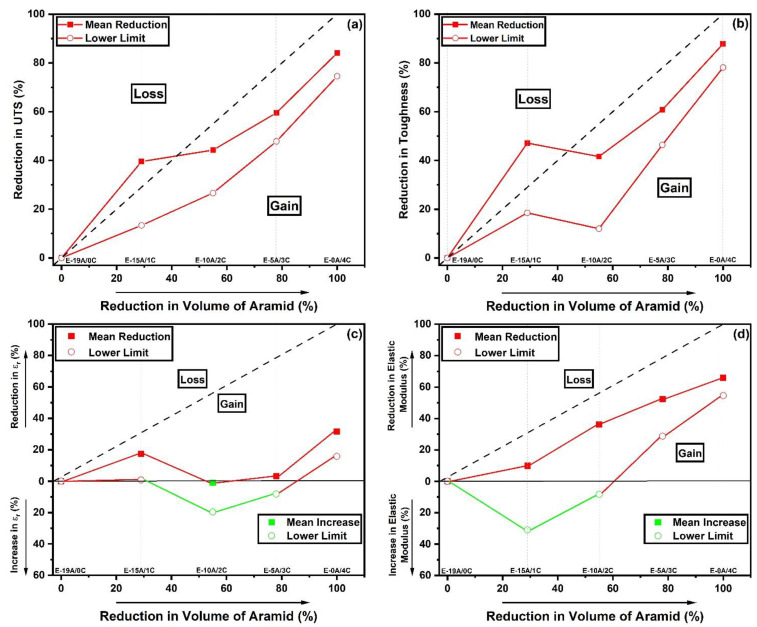
Reduction Maps for tensile properties of hybrid aramid/curaua composites: (**a**) UTS; (**b**) toughness; (**c**) deformation at rupture (ε_r_); and (**d**) elastic modulus.

**Figure 8 polymers-14-02588-f008:**
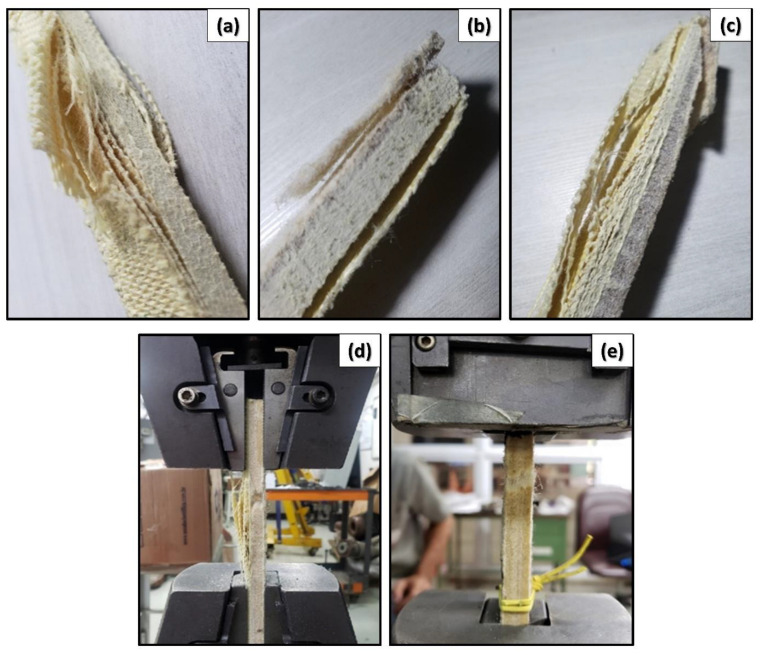
Tensile failure modes of composites: (**a**) E-19A/0C composite with delamination and yarn pullout and rupture evidences; (**b**) E-15A/1C composite with outermost aramid layer delamination and curaua fiber pullout and rupture evidences; (**c**) E-10A/2C composite with aramid layer delamination, aramid yarn rupture, and curaua fiber pullout and rupture evidences; (**d**) E-5A/3C composite suffering aramid layer delamination, aramid yarn rupture, and curaua fiber pullout and rupture; (**e**) E-0A/4C composite suffering curaua fiber pullout and rupture.

**Table 1 polymers-14-02588-t001:** Physical and mechanical properties of curaua and synthetic fibers. Adapted with permission from Ref. [16]. Copyright 2015 John Wiley & Sons, Inc.

Fiber	Density ρ (g/cm^3^)	Ultimate Tensile Strength UTS (MPa)	Elasticity Modulus E (GPa)	Ultimate Specific Strength UTS/ρ (MPa·cm^3^/g)
Curaua	0.57–0.92	117–3000	27–80	3261
E-glass	2.50–2.58	2000–3450	70–73	1380
Carbon	1.78–1.81	2500–6350	230–400	3567
Aramid	1.44	3000–4100	63–131	2847

**Table 2 polymers-14-02588-t002:** Composites configuration based on the number of reinforcing layers.

Epoxy (E) Composite Configuration	Number of Layers	
	Aramid Fabric (A) Woven Fabric	Curaua Fiber (C) Non-Woven Mat
E-19A/0C	19	0
E-15A/1C	15	1
E-10A/2C	10	2
E-5A/3C	5	3
E-0A/4C	0	4

**Table 3 polymers-14-02588-t003:** Composites’ quantitative characteristics associated with distinct layers configuration.

Composite Configuration	Weight (g)	Thickness (mm)	Vol% Total Reinforcement	Vol% Aramid	Vol% Curaua	Arealv Density (kg/m^2^)
PASGT-based	~200	~8–10	~70	-	-	11.24
E-19A/0C	197.78 ± 1.41	8.32 ± 0.03	73.29 ± 0.58	73.29 ± 0.58	0	10.99 ± 0.07
E-15A/1C	204.65 ± 0.92	9.06 ± 0.04	68.52 ± 0.63	52.06 ± 0.39	16.47 ± 0.44	11.37 ± 0.51
E-10A/2C	200.47 ± 2.83	9.37 ± 0.08	65.00 ± 0.83	33.27 ± 0.40	31.73 ± 0.763	11.14 ± 0.16
E-5A/3C	196.14 ± 2.17	9.61 ± 0.13	61.22 ± 0.59	15.96 ± 0.15	45.26 ± 0.72	10.90 ± 0.12
E-0A/4C	192.41 ± 2.21	9.98 ± 0.12	57.97 ± 0.86	0	57.97 ± 0.86	10.69 ± 0.12

**Table 4 polymers-14-02588-t004:** Composite mechanical properties obtained from tensile test.

Composite	Aramid Volume Reduction (%)	Ultimate Tensile Strength UTS (MPa)	Deformationat Rupture ε_r_ (%)	Elastic Modulus (GPa)	Toughness (MJ/m^3^)
E-19A/0C	0	280.21 ± 53.79	3.97 ± 0.30	11.40 ± 1.93	5.41 ± 1.25
E-15A/1C	29	169.49 ± 26.71	3.28 ± 0.36	10.27 ± 2.18	2.86 ± 0.53
E-10A/2C	55	156.13 ± 9.93	4.01 ± 0.38	7.26 ± 1.42	3.16 ± 0.50
E-5A/3C	78	113.32 ± 4.81	3.84 ± 0.13	5.42 ± 1.33	2.12 ± 0.11
E-0A/4C	100	44.48 ± 13.11	2.71 ± 0.38	3.87 ± 0.43	0.66 ± 0.25
E-0A/4C*	-	32.04 ± 1.44	2.38 ± 0.11	3.70 ± 0.38	0.43 ± 0.02
E-0A/4C**	-	56.92 ± 2.20	3.04 ± 0.20	4.05 ± 0.43	0.89 ± 0.06

**Table 5 polymers-14-02588-t005:** ANOVA and Tukey’s test statistical parameters for composite UTS.

Mean Treatment Squares	Mean Residue Squares	F (Calculated)	F_critic_ (Tabulated ^1^)	q (Tabulated ^2^)	MSD
89,444.46	780.08	114.66	2.54	3.99	32.21

^1^ Snedecor’s F distribution with 5% significance. ^2^ Student’s *t* distribution with 5% significance.

**Table 6 polymers-14-02588-t006:** ANOVA and Tukey’s test statistical parameters for composites deformations at rupture.

Mean Treatment Squares	Mean Residue Squares	F (Calculated)	F_critic_ (Tabulated ^1^)	q (Tabulated ^2^)	MSD
3.75	0.10	35.71	2.54	3.99	0.37

^1^ Snedecor’s F distribution with 5% significance. ^2^ Student’s *t* distribution with 5% significance.

**Table 7 polymers-14-02588-t007:** ANOVA and Tukey’s test statistical parameters for composite toughness.

Mean Treatment Squares	Mean Residue Squares	F (Calculated)	F_critic_ (Tabulated ^1^)	q (Tabulated ^2^)	MSD
35.94	0.43	82.81	2.54	3.99	0.76

^1^ Snedecor’s F distribution with 5% significance. ^2^ Student’s *t* distribution with 5% significance.

**Table 8 polymers-14-02588-t008:** ANOVA and Tukey’s test statistical parameters for composites elastic modulus.

**Mean** **Treatment Squares**	**Mean** **Residue Squares**	**F** **(Calculated)**	**F_critic_** **(Tabulated ^1^)**	**q** **(Tabulated ^2^)**	**MSD**
121.02	2.49	48.60	2.54	3.99	1.82

^1^ Snedecor’s F distribution with 5% significance. ^2^ Student’s *t* distribution with 5% significance.

**Table 9 polymers-14-02588-t009:** Charpy impact results of absorbed energy by the investigated composites.

Composite	Absorbed Impact Energy (kJ/m)
E-19A/0C	10.32
E-15A/1C	9.72
E-10A/2C	6.29
E-5A/EC	3.86
E-0A/4C	1.34

## Data Availability

The data presented in this study are available on request from the corresponding author.

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
