# Peer review of "Tensile Properties of Curaua–Aramid Hybrid Laminated Composites for Ballistic Helmet"

_polymers, 2022, doi:10.3390/polym14132588_

Round 1

Reviewer 1 Report

This paper investigated the tensile properties of curaua-aramid hybrid laminated Composites for Ballistic Helmet. The tensile properties of the hybrid composites were tested by different preparation methods. However, some necessary information is missing or incomplete. The following major comments can further improve the quality of the paper.

#1 In the abstract, too much content is used to introduce the research background, and the current research work should be described in detail. Please provide some quantitative analysis results on the mechanical properties of the hybrid plate. In addition, the hybrid mechanism of the two fibers and the influence mechanism on the mechanical properties of hybrid composite plates should be proposed.

2# It is suggested to modify or delete part of the research background in the introduction, which is the same with the abstract.

3# The introduction part is lack of two-part necessary contents. Firstly, there is a lack of general introduction to fiber-reinforced polymer composites, such as performance advantages (light weight, high strength, better mechanical properties, durability, fatigue resistance, etc.), composition (fiber, matrix and interface), classification (natural fiber and synthetic fiber composites). After the basic introduction, the authors should analyze the respective advantages of synthetic fiber and natural fiber, and further put forward the hybrid necessity of natural fiber and synthetic fiber in terms of cost, properties etc. On the other hand, this paper lacks of the research summary of the hybrid mechanism, hybrid effect of the two fibers and the effect of on the mechanical properties of hybrid composites. For the above two parts, please see the latest research on hybrid composites to make necessary supplements. https://doi.org/10.1080/15376494.2021.1974620. Materials & Design, 2019, 183: 108112. Construction and Building Materials, 2022, 315: 125710.

4# How to solve the greater hydrophilicity of natural fiber? In addition, how to consider improving the interfacial bonding properties of the two fibers and resins? In the last two parts of the introduction, the authors should put forward the effects of hybrid mechanism and hybrid effect of natural fiber and synthetic fiber on composites.

5# In the part 2.1, how about the fluidity of epoxy resin? As can be seen from Figure 2, the permeability of curaua non-woven mat seems not good, so can the epoxy infiltrate the mat well?

6# Please provide the selection basis in Table 2 for composites configuration. In addition, it can be seen in Figure 3 that the two fiber layers are arranged according to the simple way of upper and lower layers, which is easy to produce a weak interface layer, resulting in the first cracking, and even final failure of the composite in the interface layer. For the above arrangement, how do the authors consider to give full play to the excellent properties of the two materials? Why are other arrangements not considered, such as alternating arrangements?

7# In Figure 3, the authors use the hand lay-up method to prepare hybrid composite plates. How to ensure that the bubbles in the epoxy is effectively removed during the resin curing process? Generally speaking, the method of vacuum infusion is more effective than hand lay-up method to prepare composite plates.

8# Please indicate what method is used to anchor the plate during the tension process. In addition, please indicate whether the deformation or strain is obtained through extensometer or strain gauge?

9# Why only do the tensile test? After the tensile test, some other micro performance tests are very effective to expose the hybrid mechanism and effect of hybrid composite. In addition, why the impact resistance test is not considered in the current research, because the hybrid composites developed in this paper are used in the application of impact resistance.

10# Some key information about hybrid effect can be obtained from the stress-strain curves in Figure 5. However, it is very regrettable that this is not presented in the current paper.

11# Figures 6 to 9 are duplicate with the data in table 4, so it is recommended to retain only one.

12# From Figures 6 to 9, according to the change of property parameters, please summarize an optimal composites configuration.

13# The data represented in table 9 are actually the same with table 4 in different ways. Please delete one data.

14# About the analysis of tensile rupture mechanisms in part 3.3, the current discussion and analysis is not convincing only through the observation of fracture mode with the naked eye. Some key characterization and microanalysis methods were missing. It is suggested that the authors refer to the relevant literature on hybrid effect and mechanism to support the current discussion. Some scholars have conducted in-depth research and summary on the hybrid mode, hybrid effect and hybrid mechanism through the tensile properties of hybrid composite, such as Gupta M K and Srivastava R K; Xian GJ and Guo R; Rahman, R, Syed Zhafer Firdaus Syed Putra.

Author Response

Dear reviewer,

The pdf with the Response to Reviewers is attached. Thank you for your suggestions throughout the work. We believe that all the notes have contributed positively to the manuscript.

Reviewer 2 Report

Reviewers' comments:

Manuscript number: polymers-1756206

Title: Tensile Properties of Curaua-Aramid Hybrid Laminated Composites for Ballistic Helmet.

The manuscript needs a detailed editing. It cannot be recommended for publication in the present form. I hope the following points would be helpful for the authors.

The authors need to consider the following comments

- In the Abstract, the authors need to improve with more specific short results and conclusions, i.e. academic novelty or technical advantages. Furthermore, they should add the Graphical Abstract.

- 2 2.3. Uniaxial Tensile Test - should be provide more details.  

- Figure 5. Typical stress-strain (σ-ε) curves obtained from the composites tensile tests: (a) E-19A/0C, (b) E-206 15A/1C, (c) E-10A/2C, (d) E-5A/3C and (e) E-0A/4C. – Not clear make clear.

- The Results and Discussion section should be detailed especially for the 3.2 Reduction Maps.

- The authors are obliged to repeat the discussion part of the Tensile Rupture Mechanisms.

- Summary and Conclusions should be concise.

- References: make all references in same format for volume number, page number and journal name, because it is difficult to searching and reading.

Based on these, I advise the authors to rectify the above mentioned errors and we hope to re-evaluate the revised manuscript.

Author Response

Dear reviewer,

The pdf with the Response to Reviewers is attached. Thank you for your suggestions throughout the work. We believe that all the notes have contributed positively to the manuscript

Round 2

Reviewer 1 Report

The authors have addressed most of the comments. It is suggested to accept the paper.

Reviewer 2 Report

The authors have improved the revised manuscript significantly, I recommend acceptance.